# Luteolin and Vernodalol as Bioactive Compounds of Leaf and Root *Vernonia amygdalina* Extracts: Effects on α-Glucosidase, Glycation, ROS, Cell Viability, and In Silico ADMET Parameters

**DOI:** 10.3390/pharmaceutics15051541

**Published:** 2023-05-19

**Authors:** Francine Medjiofack Djeujo, Valentina Stablum, Elisa Pangrazzi, Eugenio Ragazzi, Guglielmina Froldi

**Affiliations:** Department of Pharmaceutical and Pharmacological Sciences, University of Padova, 35131 Padova, Italy; francine.medjiofackdjeujo@phd.unipd.it (F.M.D.); etwvalentina@gmail.com (V.S.); elisapangrazzi99@gmail.com (E.P.); eugenio.ragazzi@unipd.it (E.R.)

**Keywords:** flavonoids, terpenes, ORAC assay, MTT assay, antioxidants, phytotherapy

## Abstract

The aqueous decoctions of *Vernonia amygdalina* (VA) leaves and roots are widely used in traditional African medicine as an antidiabetic remedy. The amount of luteolin and vernodalol in leaf and root extracts was detected, and their role was studied regarding α-glucosidase activity, bovine serum albumin glycation (BSA), reactive oxygen species (ROS) formation, and cell viability, together with in silico absorption, distribution, metabolism, excretion, and toxicity (ADMET) properties. Vernodalol did not affect α-glucosidase activity, whereas luteolin did. Furthermore, luteolin inhibited the formation of advanced glycation end products (AGEs) in a concentration-dependent manner, whereas vernodalol did not reduce it. Additionally, luteolin exhibited high antiradical activity, while vernodalol demonstrated a lower scavenger effect, although similar to that of ascorbic acid. Both luteolin and vernodalol inhibited HT-29 cell viability, showing a half-maximum inhibitory concentration (IC_50_) of 22.2 µM (−Log IC_50_ = 4.65 ± 0.05) and 5.7 µM (−Log IC_50_ = 5.24 ± 0.16), respectively. Finally, an in silico ADMET study showed that both compounds are suitable candidates as drugs, with appropriate pharmacokinetics. This research underlines for the first time the greater presence of vernodalol in VA roots compared to leaves, while luteolin is prevalent in the latter, suggesting that the former could be used as a natural source of vernodalol. Consequently, root extracts could be proposed for vernodalol-dependent antiproliferative activity, while leaf extracts could be suggested for luteolin-dependent effects, such as antioxidant and antidiabetic effects.

## 1. Introduction

Diabetes mellitus (DM) and persistent hyperglycemia are associated with macrovascular and microvascular complications, resulting in neuropathy, nephropathy, heart disease, stroke, and vascular diseases [1]. The serious impairments associated with DM and its constant increase in the world drive researchers to pursue new antidiabetic agents, also of natural origin, expecting cheaper and safer drugs. Plant-derived medicines have always played an important role in the culture and health traditions of the world population, and are still of great interest [2]. *Vernonia amygdalina* Del. (Asteraceae), also known as ‘bitter leaf’ or ‘ndolé’ due to the bitter taste of the leaves, is an African and Asian medicinal plant well known for its use in traditional medicine in various diseases, including wound healing, malaria, hypertension, and DM [3,4,5]. Ethanol and aqueous *Vernonia amygdalina* (VA) extracts and their in vitro activity were studied as potential agents capable of counteracting hyperglycemia and the production of advanced glycation end products (AGEs), showing their activity against α-glucosidase activity and AGEs [6]. Furthermore, other authors have reported antidiabetic effects in diabetic rat models [7,8,9,10,11,12,13]. In addition to the use of VA leaves in African folk medicine, leaf preparations are also marketed as food supplements. Recently, authors have suggested its use in the preparation of beverages such as non-alcoholic wheat beers, using a different percentage of bitter leaf, suggesting VA as a dietary intervention for the treatment of DM [14].

Studies have revealed the presence in VA tissues of several phytochemicals, such as polyphenols, alkaloids, saponins, and steroid glycosides [6,15,16]. Although flavonoids are common compounds in many types of plants, which are also used for food, terpene-derived compounds are less documented. Therefore, the current study focused on the study of two representative compounds of VA, the flavonoid luteolin and the terpene vernodalol (Figure 1), assuming these to be characteristic phytoconstituents of leaf and root extracts [6].

Luteolin is a yellow flavonoid well-known for its antioxidant, anti-inflammatory, and potential anticancer activities [17,18,19]. Luteolin is very common in fruits and vegetables for food use, but it is also found in many medicinal plants, among those VA [6,16,20]. Luteolin has anti-inflammatory (through the inhibition of NF-κB, ERK1/2, MAPK, JNK, IL-6, IL-8, and TNF-α) and antioxidant (reducing reactive species and increasing endogenous antioxidants) activities [19]. Luteolin might preserve intestinal epithelial barrier function through the suppression of the STAT3 signaling pathway by protein phosphoserine phosphatase-1 (SHP-1) [21]. Several studies also show its potential role as an anticancer agent, although no approved indications are recognized [22,23]. Furthermore, its use as a hypoglycemic agent in DM was suggested in both in vitro and in vivo experimental models [20,24].

Vernodalol is a sesquiterpene lactone isolated from the roots and leaves of VA and other *Vernonia* species [6,25,26,27], and also from the seeds of *Chenopodium anthelminticum* and *Distephanus angulifolius* [28]. It shows several activities, including antileishmaniasis, insecticidal, and antimicrobial properties against bacteria and fungi [5,29,30,31]. Furthermore, antiproliferative activity has been reported in various cell lines, such as acute promyelocytic leukemia (APL), by inducing apoptosis through the mitochondrial pathway and causing cell cycle arrest in the G2/M phase [32], and in diffuse large B-cell lymphoma (DLBCL) cells, by enhancing apoptosis induced with TRAIL (tumor necrosis factor (TNF) related apoptosis-inducing ligand) [25]. Recently, in silico studies of various sesquiterpene lactones, including vernodalol, on the epidermal growth factor receptor (EGFR) and vascular endothelial growth factor receptor (VEGFR) showed their potential anticancer activity [33]. Recently, Sinisi et al. showed that vernodalol is an activator of Nrf2 (erythroid-2 nuclear transcription factor, EC_50_ = 3.2 µM), which increases the expression of the antioxidant response element-dependent gene (ARE) and therefore counteracts oxidative stress [26]. Furthermore, vernodalol inhibits STAT3/NF-κB pathways that suppress inflammatory damage [26].

Up to now, many plants have been proposed for the treatment of DM. Recently, various authors have indicated that there are several hundreds of experimentally proven medicinal plants with antidiabetic properties, but the complete mechanism has been investigated for only a hundred of them [34,35]. An ongoing challenge is to deepen the knowledge about the various mechanisms by which the antidiabetic activity of traditional herbal remedies is exploited, and, in this context, attention was specifically focused on VA.

In this research, vernodalol and luteolin are proposed as reference compounds of VA extracts and, for this, were quantified in leaf and root extracts. Therefore, both compounds were tested for their antioxidant, anti-α-glucosidase and antiglycant activities, and for their influence on HT-29 cell viability. Furthermore, in silico absorption, distribution, metabolism, excretion, and toxicity (ADMET) parameters of luteolin and vernodalol were evaluated. According to the purpose of this investigation, the assays considered were selected from the documented and available ones. It was not a screening approach, but a targeted investigation of the main mechanisms to explain the antidiabetic properties attributed to VA, to rationalize the traditional use and pose the basis for possible drug development.

## 2. Materials and Methods

### 2.1. Reagents

Acarbose, acetic acid, 2,2′-azobis(2-amidinopropane) dihydrochloride (AAPH), bovine serum albumin (BSA), dichlorofluorescin diacetate (DCFH-DA), dimethylsulfoxide, fluorescein, α-glucosidase (EC 3.2.1.20, *Saccharomyces cerevisiae* type I, ≥10 Units/mg protein), 6-hydroxy-2,5,7,8-tetramethylchroman-2-carboxylic acid (trolox), luteolin, methanol, *p*-nitrophenyl-α-D-glucopyranoside (*p*-NPG), thiazolyl tetrazolium bromide, and vernodalol were purchased from Merck Life Science S.r.l. Milan, Italy. The purity of the standards was ≥97%, while for the other compounds it was at least of analytical grade. Specifically, the purity of luteolin was 98% and that of vernodalol was 95%.

### 2.2. Vernonia amygdalina Extracts

The methods used to obtain leaf and root extracts have already been described [6]. Briefly, the tissues of spontaneous VA plants were harvested in October 2018, in Loum (Cameroon). Dr Victor Djeujo (Loum, Cameroon) confirmed the plant samples (VAL01 and VAR01). Roots (10 g) and leaves (10 g) were macerated with water (100 mL) or 70% w/w ethanol (100 mL) for 24 h. Soxhlet extracts were obtained with a Soxhlet Buchi extractor B11 containing 100 mL of water. The aqueous extracts were freeze-dried, while the ethanol extracts were dried in a water bath until they were dry.

In total, six extracts were obtained, which were: (a) aqueous root extract (AquRE), (b) aqueous leaf extract (AquLE), (c) Soxhlet root extract (SoxRE), (d) Soxhlet leaf extract (SoxLE), (e) ethanol root extract (EthRE), and (f) ethanol leaf extract (EthLE).

### 2.3. HPLC LC/DAD Analysis

High performance liquid chromatography (HPLC) analysis was carried out with an HPLC system (Waters Corporation, Milford, MA, USA) consisting of binary HPLC pumps and a photodiode array detector (DAD, Waters 2998) that collected UV spectra in the range of 210–400 nm. Selected wavelengths used for quantitative detections were 350 nm for luteolin, and 325 nm for vernodalol. Chromatographic separation was performed with a Simmetry^®^ RP C18 column, 4.6 × 75 mm, 3.5 μm column (Waters Corporation, Milford, MA, USA). A gradient elution program consisted of phase A (water 0.1% acetic acid) and phase B (methanol 0.1% acetic acid). The flow rate was 1 mL/min. The separations of the VA extracts were obtained by injecting 20 µL of 1 mg/mL of each extract, while the gradient profile was as follows: 1 min, 95% A; 7 min, 75% A; 9 min, 60% A; 12 min, 55% A; 14 min, 50% A; 18 min, 40% A; 23 min, 20% A; 28 min, 95% A; 30 min, 95% A. Before injection, the methanolic solution of each extract was filtered through a 0.45 µm membrane filter.

Each compound was identified by the use of the retention time, co-injection, and spectral accordance with the luteolin and vernodalol standards.

### 2.4. Oxygen Radical Absorbance Capacity (ORAC) Assay

The ORAC assay evaluates the capacity of substances to delay oxidative reactions caused by peroxyl radicals through the hydrogen atom transfer (HAT) mechanism [36]. The assay was conducted as previously described [37]. In short, trolox solution was prepared in phosphate buffer at four different concentrations, from 6.25 to 50 µM. In 24-well plates, 1.5 mL of fluorescein (0.08 µM) was added to 250 µL of trolox (controls), or 250 µL of buffer (blanks), or 250 µL of each tested compound (treatment). After the incubation (37 °C, 10 min), 250 µL of 0.15 M AAPH was added. Subsequently, with a microplate reader (PerkinElmer VICTOR^®^ Nivo™ Waltham, MA, USA), a kinetic fluorescence reading was performed at 37 °C for 45 min, at the wavelengths 485 nm (excitation) and 530 nm (emission). The final values were reported in TEAC (trolox equivalent antioxidant capacity, µmol TE/µmol compound).

### 2.5. α-Glucosidase Inhibition Assay

The capacity of luteolin and vernodalol to reduce the activity of α-glucosidase (*Saccharomyces cerevisiae* type I lyophilized powder) using the substrate *p*-nitrophenyl-α-D-glucopyranoside (pNPG) [38]) was tested. As a positive control, acarbose was used. Samples were incubated for 10 min, at 37 °C, with α-glucosidase 0.05 µM and 0.1 M PBS (pH 6.8). The reaction started with the addition of 4 mM pNPG. Absorbance values at 405 nm were detected for 45 min, using a PerkinElmer VICTOR^®^ Nivo™ microplate reader spectrophotometer (Waltham, MA, USA).

### 2.6. Protein Glycation Inhibition

Protein glycation was obtained in accordance with a previously described method [39,40]. AGEs were detected using BSA (10 mg/mL, pH 7.4) as the protein substrate, and glucose (1.0 M), fructose (0.5 M), or ribose (0.05 M) as glycation inducers. Each compound was tested from 10 to 100 µM by adding the substrate, incubating at 37 °C for 14 days. The intensity of fluorescence was detected at 355 nm (excitation wavelength) and 460 nm (emission wavelength) by a PerkinElmer Victor Nivo microplate reader (Waltham, MA, USA). The reduction in fluorescence in the presence of each compound was calculated as the difference in fluorescence between the maximum fluorescence (control) and the fluorescence in the presence of the tested compound. The positive control was aminoguanidine (2.5 mM, AG) [41].

### 2.7. Cell Viability Assay

Cellular proliferation was assessed with MTT assay [42]. The human Caucasian colon adenocarcinoma cell line (HT-29, Merck, Darmstadt, Germany) was grown in RPMI-1640 medium with 10% fetal bovine serum, 2% penicillin and streptomycin (Merck, Darmstadt, Germany) in sterile flasks, kept in incubator at 37 °C (atmosphere 5% CO_2_). The cells were cultured for up to ten passages. Cells were seeded in a 96-well plate, density 5000 in each well, and allowed to grow for 24 h; then, these were treated with the tested compounds or medium (control). Compound solutions were in DMSO (<1% *v*/*v*). After 24 h, cells were treated for 4 h with 500 µg/mL MTT. Vital cells with MTT produce purple crystals of formazan salt, soluble in 2-propanol. The absorbance was detected with a PerkinElmer VICTOR^®^ Nivo™ microplate reader (Waltham, MA, USA), at λ 570 nm.

### 2.8. Prediction of Absorption, Distribution, Metabolism, Excretion, and Toxicity (ADMET) Properties

The estimated properties of ADMET were achieved with the pharmacokinetics web tool pkCSM [43]. The pharmacokinetic parameters considered were the logarithmic ratio of the partition coefficient (LogP), the ability to reach and pass the blood–brain barrier (BBB), absorption at the human intestinal level (%), volume of distribution (log L/kg), total clearance (log CL_tot_), and oral acute toxicity (LD_50_ mol/kg). The canonical SMILE molecular structures of the tested compounds were obtained from PubChem [44].

### 2.9. Statistical Analysis

Data are expressed as mean ± SEM of at least three independent experiments. Sigmoid curve fitting and statistical evaluations were performed using GraphPad Prism 9 software (San Diego, CA, USA). Statistical comparisons were performed using ANOVA, followed by Tukey’s multiple comparison test. The half-maximum inhibitory concentration (IC_50_) was obtained by nonlinear regression. The difference between control and treatment was evaluated using the Student’s *t* test. The significance level was established at *p* < 0.05.

## 3. Results and Discussion

### 3.1. Measurements of Luteolin and Vernodalol in Leaf and Root Vernonia amygdalina Extracts

In a previous in vitro investigation, ethanol and aqueous extracts of VA leaves were studied for their potential use in DM, highlighting their ability to reduce BSA glycation and α-glucosidase activity [6]. In the current research, the quantification of luteolin and vernodalol in the extracts was determined to define the role of the two compounds in plant activity, since luteolin and vernodalol can be defined as reference compounds for VA extracts. Their identification and quantification in three types of leaf extracts, i.e., macerate aqueous leaf extract (AquLE), Soxhlet aqueous leaf extract (SoxLE), and macerate ethanol leaf extract (EthLE), and three root extracts, i.e., macerate aqueous root extract (AquRE), Soxhlet aqueous root extract (SoxRE), and macerate ethanolic root extract (EthRE), were determined with HPLC analysis using the calibration lines obtained with the standards, repeating the analysis four times. Figure 2 shows illustrative chromatograms of vernodalol characterization in root extracts. EthLE and SoxLE have the highest amount of luteolin, respectively, equal to 0.54 ± 0.04 mg/g and 0.49 ± 0.04 mg/g, compared to AquLE equal to 0.21 ± 0.01 mg/g (Figure 3). On the other hand, the amount of luteolin in the root extracts was not measurable. Otherwise, the HPLC analysis performed on the root extracts of VA allowed one to highlight an elevated presence of vernodalol. The highest amount of the sesquiterpene was detected in EthRE, with a value of 336.79 ± 15.60 mg/g, compared to SoxRE (176.58 ± 11.49 mg/g) and AquRE (47.81 ± 0.70 mg/g); see Figure 3. Otherwise, the amount of vernodalol in the leaf extracts was not quantifiable. In particular, the HPLC analysis demonstrated that the amount of compound extracted varies with the extraction method, providing useful information for the optimal management of raw materials designated for medicinal herbal use. The data obtained also confirm previous research, which detected the presence of luteolin in extracts of AV leaves, supporting the VA antioxidant and cardioprotective effects [6,45,46].

### 3.2. Antiradical Activity

Antioxidant activity is a new way to deal with complications of DM [47]. Increased free radical production and increased oxidative stress are well-known recognized factors affecting the morbidity and mortality of DM [48]. Therefore, it is essential to focus on inhibitors to study new aspects of diabetes treatment that are not related to direct hyperglycemia control. For example, the presence of antioxidant activity identified in the mechanisms of action of current antidiabetic drugs, such as metformin, glibenclamide, and repaglinide [49], represents a new horizon that should deserve in-depth analysis. In general, subjects with DM are more exposed to oxidative attacks than healthy people due to their higher reactive oxygen species (ROS) production [50]. Therefore, compounds having antioxidant properties are of interest in the treatment of cardiovascular diseases and diabetes. Luteolin and vernodalol revealed a beneficial antioxidant activity detected by the ORAC assay. The TEAC values of luteolin and vernodalol were 7.34 ± 1.20 and 0.82 ± 0.11 μmol TEAC/μmol compound, respectively (Figure 4). Although the vernodalol value was lower than that of luteolin, it was similar to that of ascorbic acid (0.94 ± 0.13 μmol TEAC/μmol compound), which is generally considered a good antioxidant.

### 3.3. α-Glucosidase Activity

α-Glucosidase is an interesting pharmacological target for the treatment of hyperglycemia because the inhibition of enzyme activity in the intestinal tract can reduce the amount of glucose available for systemic uptake in the human organism. Therefore, vernodalol and luteolin were tested in vitro against the enzyme α-glucosidase. Vernodalol from 1 µM to 25 µM did not decrease the α-glucosidase activity, while luteolin reduced enzyme activity in a concentration-dependent manner, from 5 µM to 50 µM (Figure 5), with an IC_50_ of 16.2 µM (−Log IC_50_ = 4.79 ± 0.03). These luteolin results are consistent with those previously reported [24]. Although vernodalol did not show inhibitory activity, the study is interesting because it is the first to evaluate the effect of vernodalol on α-glucosidase activity.

### 3.4. Antiglycation Activity

Sustained hyperglycemia increases the formation of AGEs produced by the non-enzymatic glycation of proteins and lipids in subjects affected by DM [51]. Therefore, glycation was studied using BSA with three different sugars, such as ribose, fructose, and glucose, incubated for 7 days (ribose) or 14 days (fructose and glucose). Luteolin and vernodalol were tested as inhibitors in a concentration range of 10 µM to 100 µM (Figure 6). The glycation obtained with ribose and fructose and glucose showed different kinetics in AGE formation over time (Appendix A). According to the literature, ribose was found to be the most potent and efficient glycation agent, since fructose induced a lower BSA glycation compared to ribose, even higher than that of glucose. Luteolin inhibited the glycation of BSA induced by ribose, fructose, and glucose in a concentration-dependent manner (Figure 6). The IC_50_ values were 58 µM (−Log IC_50_ = 4.24 ± 0.02), 19.6 µM (−Log IC_50_ = 4.71 ± 0.04), and 40.4 (−Log IC50 = 4.39 ± 0.05), respectively. These results are consistent with the findings of Wu et al. who showed for luteolin an IC_50_ of 16 µM in glucose-induced BSA glycation [52]. Similarly, Muramatsu et al. reported an IC_50_ of 19.55 µM in glucose-induced human serum albumin glycation [53]. Thus, the current results consolidate previous data from the literature, showing the excellent antiglycation activity of luteolin. Otherwise, no studies have been conducted on the effect of vernodalol on glycation. In the present investigation, vernodalol did not inhibit AGE formation under any of the experimental conditions examined (Figure 6). Therefore, it can be deduced that the antiglycant effects previously reported with *Vernonia amygdalina* extracts could be attributed, at least in part, to the activity of luteolin and not to vernodalol [6].

### 3.5. HT-29 Cell Viability

The MTT assay is a widely used method to investigate cell proliferation, which is useful for defining the potential cytotoxicity of a compound, allowing the quantification of inhibitory activity on cell viability [54]. Luteolin and vernodalol were investigated at concentrations of 0.35 µM to 35 µM to detect their influence on HT-29 cells. (Figure 7). Experimental data showed that vernodalol exhibited a higher inhibition of cell growth than luteolin. In detail, luteolin and vernodalol showed an IC_50_ of 22.2 µM (−Log IC_50_ = 4.65 ± 0.05) and 5.7 µM (−Log IC_50_ = 5.24 ± 0.16), respectively. Therefore, vernodalol was approximately four times more cytotoxic than luteolin in the HT-29 cell line. Other authors studied luteolin on the viability of several types of cell lines, for example Eca109, A549, SH-SY5Y, MG-63, and HL-60 cells, respectively, obtaining IC_50_ values of 70.7 µM, 40.2 µM, 27.1 µM, 34 µM, and 18.4 µM [55,56,57,58,59]. The values are comparable to the present result, taking into account the different cell models and the various test conditions. On the other hand, few studies have been conducted on vernodalol. Wu et al. studied vernodalol in three human acute promyelocytic leukemia (APL) cell lines (NB4, KG-1a, and HL-60), obtaining IC_50_ values from 65.7 µM to 76.4 µM (24 h of incubation) [32].

### 3.6. In Silico Prediction of ADMET Properties

Computational methods for the in silico prediction of descriptors related to the pharmacokinetics and toxicity profile of small molecules provide a useful tool for the identification of drug candidates [60]. Therefore, luteolin and vernodalol were studied using pkCSM, a pharmacokinetic web tool [43]. The pharmacokinetic parameters considered were the partition coefficient, the ability to reach and cross the blood–brain barrier (BBB), the absorption at the human intestinal level, the volume of distribution, the total clearance, and the oral acute toxicity (Table 1). Luteolin is a small molecule with good pharmacokinetic characteristics, with elevated intestinal absorption (>80%) and a high volume of distribution (Table 1). Both luteolin and vernodalol are substrates of P-glycoprotein, poorly distributed in the brain, and have similar total clearance. Furthermore, the predicted oral acute toxicity was very similar (Table 1).

Few in vivo pharmacokinetic and toxicological studies are available for luteolin, and none for vernodalol. Thus, in particular for vernodalol, this in silico evaluation shows that both compounds could be developed as pharmacological agents with suitable intestinal absorption and low toxicity.

Studies carried out in rats have shown that luteolin has moderate bioavailability (26 ± 6%), undergoes enterohepatic recycling, and has an extensive metabolism by conjugation [61]. Limited in vitro investigations have studied the interaction of luteolin with CYP enzymes, showing moderate inhibitory action on CYP2C8 and CYP1A2 [62], CYP2C9, and CYP3A [63,64,65]. In general, the data suggest interest in the development of luteolin in therapy, and several drug delivery studies are ongoing [66]. By contrast, no in vivo data are available for vernodalol, and for this new studies are needed.

The in silico ADMET parameters evaluated suggest the favorable pharmacokinetics of luteolin and vernodalol, and also when they are derived from sources other than VA. Indeed, it should be noted that even in the presence of a reasonable pharmacodynamic profile, a compound lacking favorable ADMET characteristics will hardly become a drug candidate.

## 4. Conclusions

The *Vernonia* genus, and particularly the *Vernonia amygdalina* species, has reached increasing popularity as a medicinal plant due to its traditional use as an antidiabetic treatment in Africa [15], where DM has increased rapidly as a public health problem [67]. VA was found in experimental animal models to suppress gluconeogenesis and potentiate glucose oxidation [13], emerging as a candidate for the further characterization of its medicinal role. In this regard, the ethnopharmacology process also aims to identify the most relevant chemical entities involved in a given pharmacological action, and the pharmacological investigation of new potential drugs today considers traditional medicinal plants as a very promising source [68], also recently confirmed by authoritative reviews [69].

Published studies have previously documented the presence of several bioactive compounds in VA, including flavonoids, saponins, alkaloids, tannins, terpenes, and phenols [15]. The present research showed that aqueous and ethanol extracts of leaves and roots from VA contain significant amounts of luteolin and vernodalol, respectively, since the terpenoid vernodalol is the predominant compound in root extracts, while the flavonoid luteolin occurs primarily in leaf extracts. Therefore, luteolin and vernodalol can be proposed as reference compounds helpful in standardizing the medicinal preparations of VA. The study of the two isolated compounds showed that luteolin has a very high antiradical activity, about 7–8 times greater than the reference ascorbic acid, while vernodalol showed an effect comparable to that of ascorbic acid. Vernodalol did not inhibit the α-glucosidase activity and BSA glycation; conversely, luteolin induced an appreciable concentration-dependent inhibition of both α-glucosidase activity and BSA glycation. In detail, the IC_50_ of luteolin against α-glucosidase was 16.2 µM, while the IC_50_ values against ribose, fructose, and glucose-induced BSA glycation were 58.0 µM, 19.6 µM, and 40.4 µM, respectively. Therefore, the properties of luteolin support the use of VA in the treatment of diabetes mellitus in traditional medicine.

The cytotoxicity of both compounds was evaluated regarding the viability of HT-29 cells, showing for luteolin and vernodalol IC_50_ values of 22.2 and 5.7 µM, respectively. Therefore, vernodalol was a more powerful inhibitor of HT-29 cell proliferation than luteolin. The in silico pharmacokinetics and toxicity profile suggest that both compounds can be suitable candidates as drugs, having adequate pharmacokinetic behaviors.

This research indicates for the first time that vernodalol is the predominant constituent of VA root extracts compared to leaf extracts, suggesting that the roots of the plant could be used as a natural source of vernodalol. Consequently, root extracts could be proposed for vernodalol-dependent activities, such as antiproliferative activity, while leaf extracts could be proposed for luteolin (flavonoid)-dependent properties, such as antioxidant, antiglycant, and antihyperglycemic effects.

The results validate the use of VA as a local resource in African and Asian regions, in line with WHO recognition of the use of herbal remedies for the treatment of DM [2]. In Africa, this plant is commonly considered a food [15,70], in addition to its medicinal properties. This provides additional relevance to support the use of VA, since it can provide nourishment and, at the same time, due to the properties investigated, act as a prevention of the progression of metabolic diseases, such as diabetes. Therefore, this plant could be included among functional foods or nutraceuticals, another important and current research field for diabetes prevention and management [71,72]. Additionally, these data, along with those available in the literature, support the development of luteolin as a potential pharmacological agent in the prevention and treatment of diseases related to hyperglycemia, as a consequence of its antioxidant and antidiabetic effects here demonstrated. Relevant clinical trials are required to validate its therapeutic use.

## Figures and Tables

**Figure 1 pharmaceutics-15-01541-f001:**
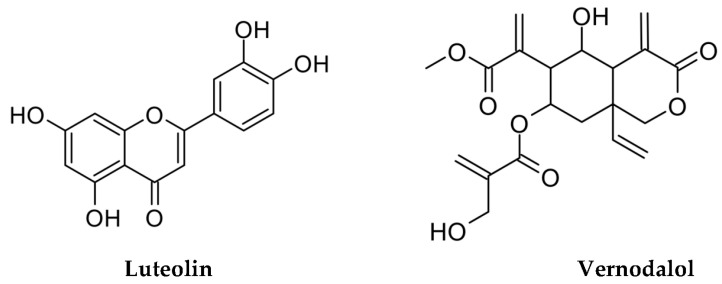
Chemical structures of two characteristic compounds detected in *Vernonia amygdalina* extracts.

**Figure 2 pharmaceutics-15-01541-f002:**
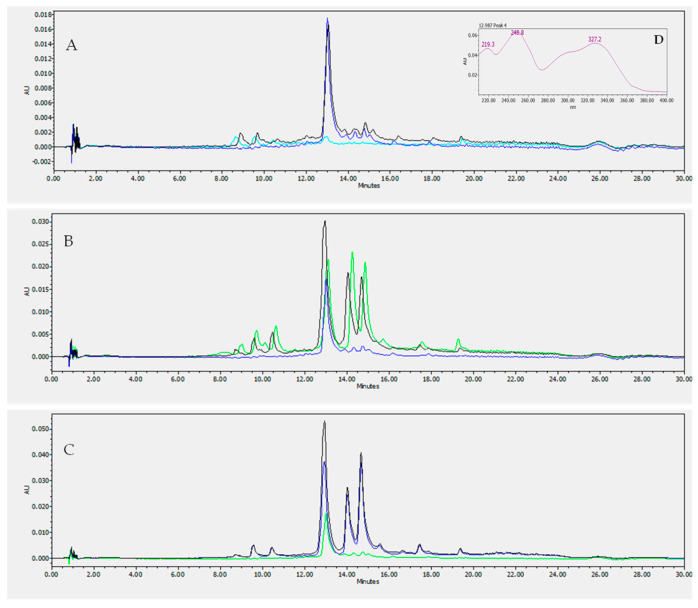
Exemplificative HPLC-DAD chromatograms: (**A**) 1 mg/mL macerate aqueous root extract (AquRE, clear blue), AquRE + 0.125 mg/mL vernodalol (black), and 0.125 mg/mL vernodalol alone (blue); (**B**) 1 mg/mL Soxhlet aqueous root extract (SoxRE, green), SoxRE + 0.125 mg/mL vernodalol (black), and 0.125 mg/mL vernodalol alone (blue); and (**C**) 1 mg/mL macerate ethanol root extract (EthRE, blue), 1 mg/mL SoxRE + 0.125 mg/mL vernodalol (black), and 0.125 mg/mL vernodalol alone (green). (**D**) The insert shows the UV absorption spectrum of vernodalol.

**Figure 3 pharmaceutics-15-01541-f003:**
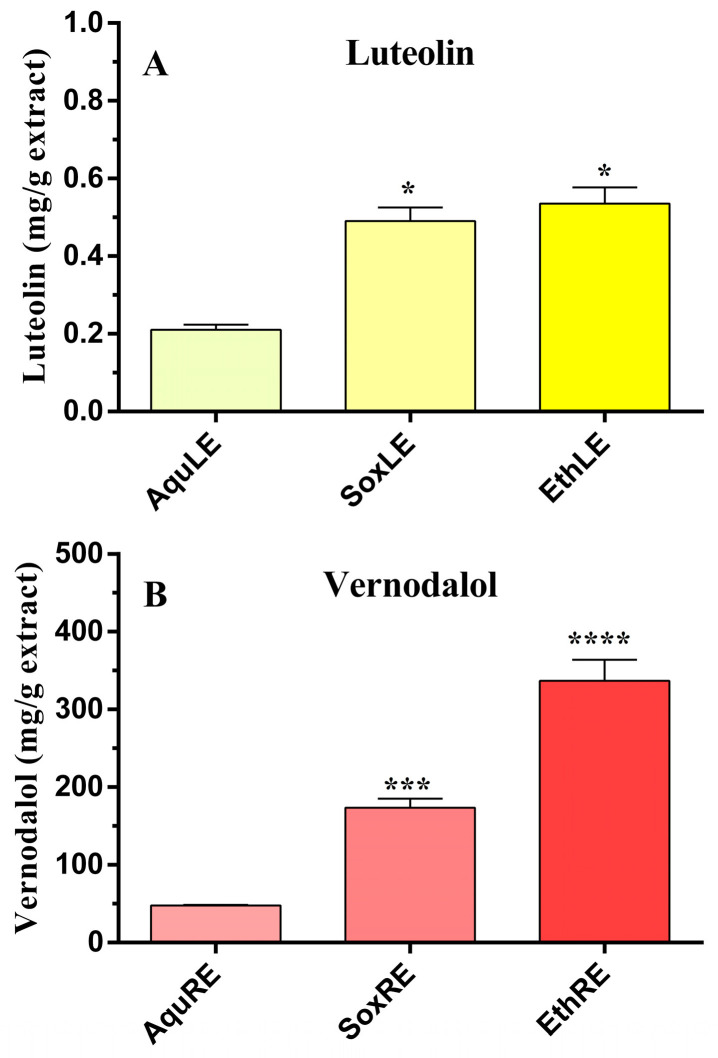
Quantitative detection of luteolin (**A**) and vernodalol (**B**) in leaf and root extracts of *Vernonia amygdalina*. Data are expressed as mean ± SEM of 4–5 experiments. AquLE, aqueous leaf extract; AquRE, aqueous root extract; SoxLE, Soxhlet aqueous leaf extract; SoxRE, Soxhlet aqueous root extract; EthLE, ethanol leaf extract and EthRE; ethanol root extract. * *p* < 0.05, *** *p* < 0.005, **** *p* < 0.0001 versus macerate aqueous extracts (AquLE or AquRE).

**Figure 4 pharmaceutics-15-01541-f004:**
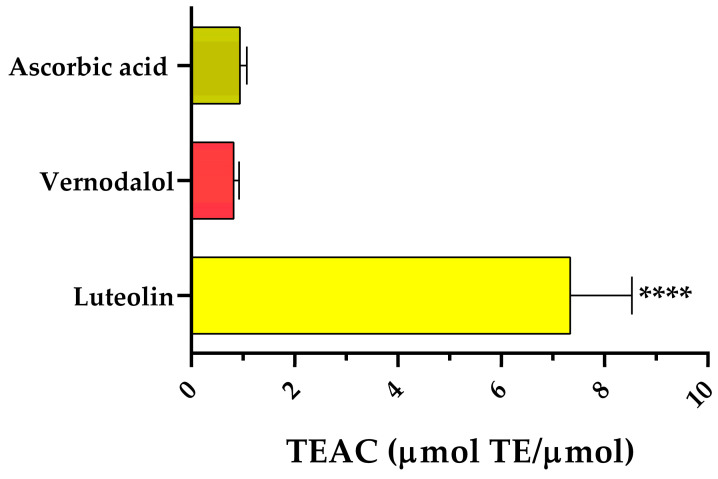
Antiradical activity of luteolin and vernodalol detected by ORAC assay. Positive control: ascorbic acid. Data are the mean ± SEM of 3–6 experiments. **** *p* < 0.0001 versus positive control (ascorbic acid). TEAC: trolox equivalent antioxidant capacity.

**Figure 5 pharmaceutics-15-01541-f005:**
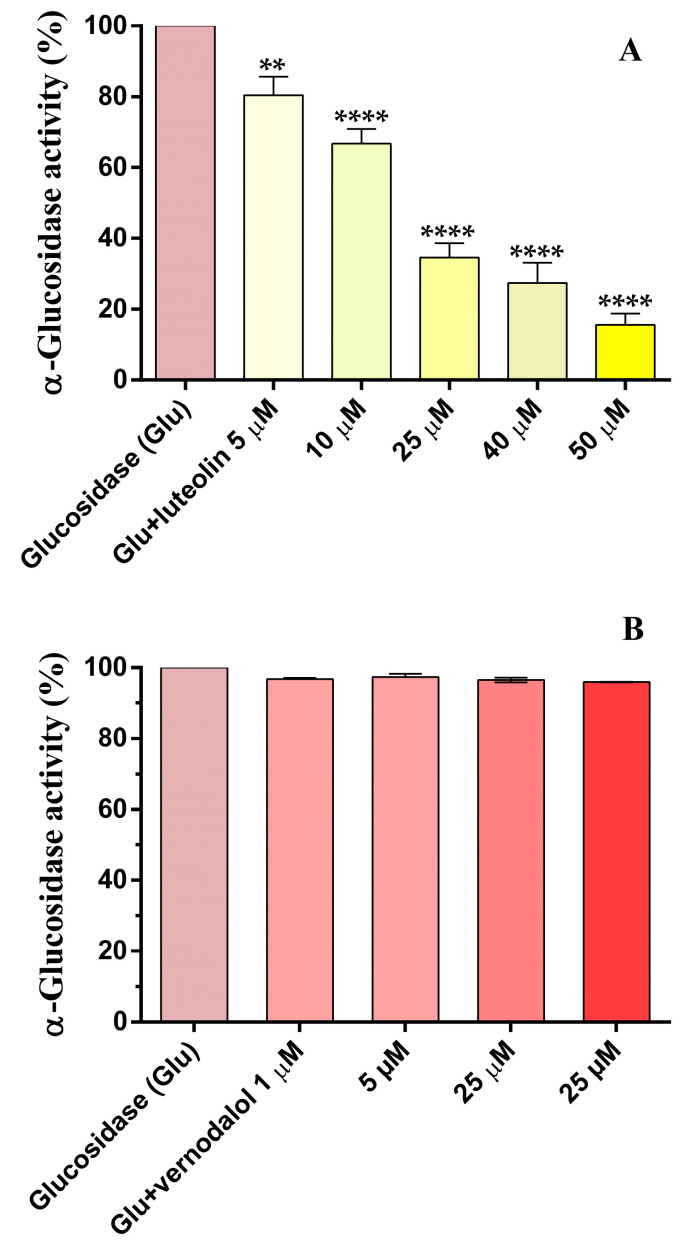
Effects of luteolin (**A**) and vernodalol (**B**) on α-glucosidase activity. The activity is expressed in percentage of the enzymatic action without inhibitor. Vernodalol was tested only up to 25 µM because no inhibition was observed. Data are the mean ± SEM of 4–6 experiments. **: *p* < 0.01; **** *p* < 0.0001 versus α-glucosidase activity.

**Figure 6 pharmaceutics-15-01541-f006:**
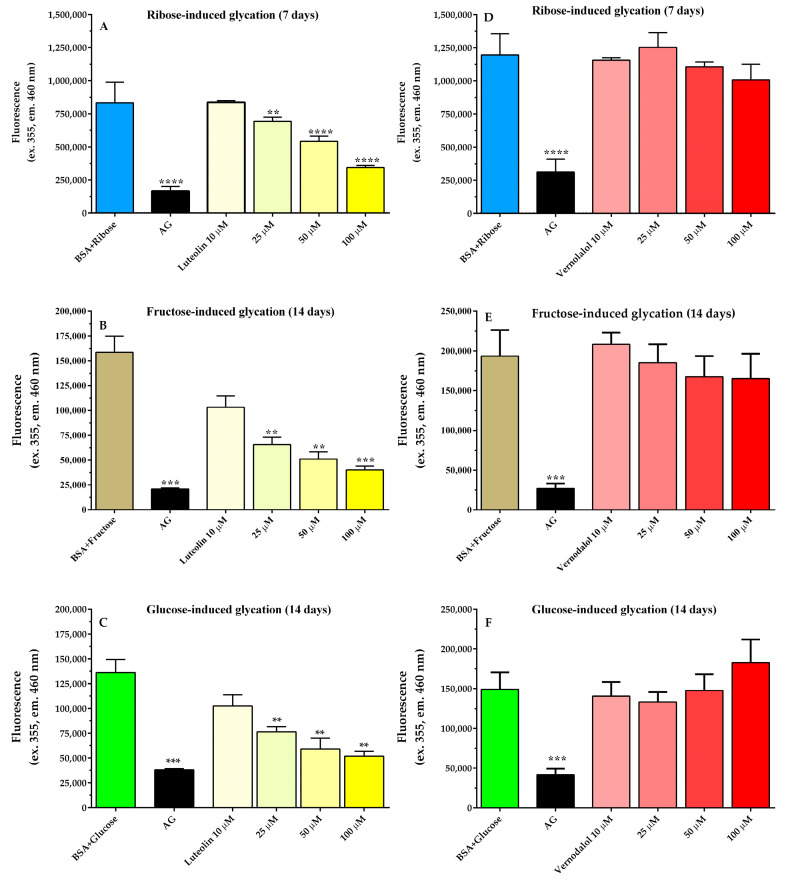
Effects of luteolin (yellow bars) and vernodalol (red bars) on AGE formation after 7 (**A**,**D**) or 14 (**B**,**C**,**E**,**F**) days of incubation of 10 mg/mL BSA with 0.05 M ribose, 0.5 M fructose (**B**), or 1.0 M glucose (**C**). Aminoguanidine (2.5 mM, AG) was the positive control. Data are the mean ± SEM of 3–6 experiments. ** *p* < 0.01, *** *p* < 0.001, **** *p* < 0.0001 versus BSA glycation (BSA + glycation agent).

**Figure 7 pharmaceutics-15-01541-f007:**
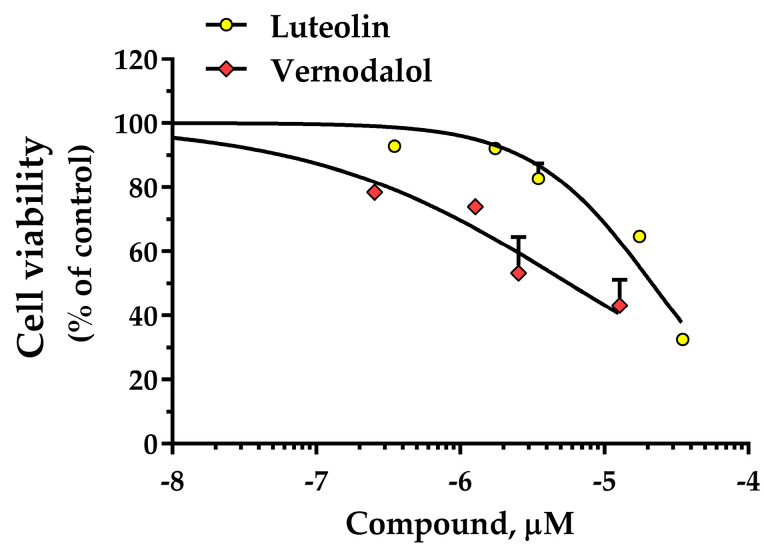
Effects of luteolin and vernodalol on the viability of human colon adenocarcinoma (HT-29) cells expressed as percentage of control. Data are the mean ± SEM of 3–6 experiments.

**Table 1 pharmaceutics-15-01541-t001:** In silico pharmacokinetic parameters of luteolin and vernodalol.

Compound	MW (g/mol)	LogP	BBB Permeability	P-Glycoprotein Substrate	CYP Substrate	Intestinal Absorption (Human) %	VD_ss_ (Human) Log L/kg	CL_tot_ (Log mL/min/kg)	Oral Acute Toxicity (LD_50_)
Luteolin (flavone)	286.239	2.2824	−0.907	Yes	CYP1A2 CYP2C6	81.13	1.153	0.495	2.45 mol/kg rat
Vernodalol (terpene)	392.404	0.4583	−0.48	Yes	No	75.39	−0.197	0.747	2.39 mol/kg rat

MW: Molecular weight, LogP: logarithmic ratio of partition coefficient, BBB: blood–brain barrier, CYP: human cytochrome P450, VD_ss_: volume of distribution, CL_tot_: total clearance (hepatic and renal clearance). The pharmacokinetic parameters were obtained using the pkCSM platform [43].

## Data Availability

All results are contained in the article and Appendix A.

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
