# Peer review of "Luteolin and Vernodalol as Bioactive Compounds of Leaf and Root *Vernonia amygdalina* Extracts: Effects on α-Glucosidase, Glycation, ROS, Cell Viability, and In Silico ADMET Parameters"

_pharmaceutics, 2023, doi:10.3390/pharmaceutics15051541_

Round 1

Reviewer 1 Report

In the manuscript entitled “Luteolin and Vernodalol as biomarkers of leaf and root Vernonia amygdalina extracts: effects on α-glucosidase, glycation, ROS, cell viability and in silico ADMET parameters”, the author investigated luteolin and vernodalol obtained from aqueous, Soxhlet and ethanol extracts of leaves and roots of Vernonia amygdalina herb. For characterisation, amount determination, antioxidative and biological activity of the examined compounds they used several experimental approaches, as well as a theoretical approach for predicting the pharmacokinetic activity. Obtained results authors compared with the results of some other authors published in the appropriately cited literature.

The writing style is very clear and concise. The results are interesting and presented correctly.

With everything mentioned in mind, I suggest publishing this review article after minor technical changes:

1.       The abbreviation ADMET should be explained. Moreover, the discussion and significance of the results obtained by the in silico method should be improved, and written, presented, and described in more detail.

2.       The English language should be checked and corrected.

3.       Spacing in the text is not the same everywhere, and it should be uniform.

Author Response

Authors’ answers

The authors thank the Reviewer for the evaluation of the article and the positive assessment.

  1. The acronym ADMET is well known in pharmacological field; thus, it was not explained in the abstract. However, the abbreviation ADMET was explained at its first mention in the introduction. In addition, in the Results section, a sentence was added highlighting the importance of the in silico study in the study of pharmacological agents.
  2. English was carefully revised.
  3. The spaces and editorial format have been revised according to the suggestions of the Reviewer.

The authors are grateful to the Reviewer for the recommendations that helped to improve the manuscript

Reviewer 2 Report

About the article entitled: Luteolin and Vernodalol as biomarkers of leaf and root Vernonia amygdalina extracts: effects on α-glucosidase, glycation, ROS, cell viability and in silico ADMET parameters, I follow with the following recommendations:

- the title of the work could infer more about the general pharmacological activity investigated, regarding the objective of the work;

- could the summary indicate the hypothesis studied, the results found indicate that the compounds are antiproliferative, antioxidants, and how is this important for the course of diseases?

- The introduction already mentions diabetes in the first line, I suggest that authors consider the information for the title and abstract;

- Regarding item 2.7, what is the passage of the cells used?

- about conclusion, I suggest the authors be more concise in their statements;

- figure 2: is it possible to leave all the figures with the same scale?

  • I indicate to the authors that they reorganize the abstract, considering the previous notes. The title can be improved to better reflect the study findings. It is a screening, with relevant results, but it is worth noting that new tests must be conducted to better support the results presented.

Author Response

The authors thank the Reviewer for comments and suggestions that helped to improve the manuscript.

- The title was changed to “Luteolin and Vernodalol as bioactive compounds of leaf and root Vernonia amygdalina extracts: effects on α-glucosidase, glycation, ROS, cell viability and in silico ADMET parameters” to make it more relevant to the results obtained from the research carried out. Thanks for the suggestion.

- The journal "Pharmaceutics" requires a short abstract of about 200 words at maximum. Taking into account the suggestions of both the Referees and the Journal indications, the abstract was shortened and the text was revised to better understand the results obtained with the research carried out.

- The proposed activity of Vernonia amygdalina as an antidiabetic remedy has been added to the abstract to support the research presented.

- Section 2.7. In the MTT assay, cells were cultured up to 10 passages. The information was also added in the Methods section.

- As regards the Conclusions section, we have tried to summarize the most significant results, without going into excessive details that might distract the reader from main findings. Unfortunately, another reviewer suggested extending this part of the manuscript. Based on the various types of recommendations, the conclusion was not significantly modified.

- Figure 2. This figure shows exemplificative chromatograms obtained by HPLC-DAD analysis of three different extracts of Vernonia amygdalina. The instrument automatically changes the y-axis for better recording of each peak. Therefore, if the same absorbance range is forced, some peaks would be very low and would not be visible to the operator, as for the aqueous root extract (Figure 2 A).

- The Abstract and the title of the manuscript have been reviewed. As well, the manuscript has been revised as suggested.

Thanks again for Reviewer’s comments, which helped us to increase the scientific merit of our work.

Reviewer 3 Report

This study reports about two selected compounds in leaf and root Vernonia amygdalina extracts. However, this is one of thousadns of studies suggesting/advertising some plant extracts that might exert preventive or even therapeutic activities. Consequently, this study comes quite fiddling. More specific highlighting of novel aspects would be helpful to improve this.

1) Why is this luteolin here better than from other plant sources ?

2) Why only luteolin and vernodalol ? Why not a handful of compounds or a certain profile ?

3) As abstracts are often placed away from the main text, abstract needs more storytelling.

4) What was really aimed ? Why is this good against diabetes ? Aren't there better plants ?

5) Selection of assays not well explained. Comes like a selection of everything. Why ? Screening appraoches are not a good justification for a scientific publication....

6) Why antioxidant activity ? This story becomes after thousands of studies quite boring...where are novel aspects ?

7) In my point of view, the term "biomarker" is used for something different. Please define or explain your own point of view...

8) What is the added-value of the ADMET values presented here compared to data already given for luteolin and vernodalol from other plant sources ?

9) Quality of figures is not comprehensive. Especially, Figure 1. This is just a copy or even a screenshoot (?).....

10) Is there a need/explanation for the different colors and patterns of the different bars in the figures ?

11) Discussion comes quite short. More comparisons/relations to all of the aspects are necessary. Otherwise, it comes like an advertisement....

12) In this context, 43 out of 54 literature citations were already introduced till the end of M&M... I miss the novel aspects or the highlights from the flavonoid or terpene biochemistry/phytochemistry

13) Funding: What is MIUR ? A company for selling Vernonia amygdalina ?

14) Conclusion is more like a summary. Where is the story ? Does this help Africa to get rid of cancer and diabetes and improve health and infrastructure ? Or another proof to keep those countries underdeveloped.... New pharmaceuticals from Africa in the future or just exploitation in Italy ?

15) Relations to food/food chemistry ?

Author Response

The authors thank the reviewer for the evaluation of the manuscript and for his/her comments and suggestions. All proposed issues were carefully examined, and modifications were made to the manuscript.

  1. Luteolin is always luteolin, regardless of the source from which it is extracted, obviously. Luteolin is in Vernonia amygdalina a characteristic constituent which in this research has been quantified. It may, at least in part, explain the utility of Vernonia preparations in traditional medicine. Therefore, it was determined quantitatively in the extracts and, moreover, investigated using several different assays.
  2. The Vernonia genus, and particularly the Vernonia amygdalina species, has achieved increasing popularity as a medicinal plant due to its traditional use as an antidiabetic treatment in Africa (Alara et al., 2017), where diabetes mellitus been rapidly increasing as a public health problem (Ndip et al., 2013). Since several years, the World Health Organization has recognized and welcomed the use of medicinal plants for the treatment of diabetes mellitus. (Modak et al., 2007). Vernonia amygdalina, found in experimental animal models to suppress gluconeogenesis and potentiate glucose oxidation (Atangwho et al., 2014), emerges as a candidate for further characterization of its medicinal role. Luteolin and vernodalol have been suggested as characteristic components of this plant. Therefore, the present investigation focused on the specific constituents of plant extracts of which luteolin is one of the most interesting. It should be noted that the ethnopharmacology process also aims to identify the most relevant chemical entities involved in a given pharmacological action, and today’s pharmacological investigation of new potential drugs considers traditional medicinal plants as a very promising source (Gurib-Fakim 2006), also recently confirmed by authoritative reviews (Newman & Cragg, 2020). Therefore, the authors believe that Vernonia amygdalina deserves further investigation, as well as luteolin and vernodalol, although it is known that luteolin can be found in several plants, including vegetables and fruits (Lin et al., 2008). Published studies have previously documented the presence of several bioactive compounds in Vernonia amygdalina, including flavonoids, saponins, alkaloids, tannins, terpenes, and phenols (see, e.g., Alara et al., 2017). The hypothesis of this research is that luteolin and vernodalol could be two relevant phytoconstituents, which have a role in the antidiabetic activity of the plant, as stated in the introduction. Of course, future research could consider additional compounds, improving knowledge about Vernonia amygdalina.
  • Alara et al. Phytochemical and pharmacological properties of Vernonia amygdalina: a review. JCEIB 2017; 2: 80–96. https://doi.org/10.15282/jceib.v2i1.3871
  • Ndip, R. N., Tanih, N. F., & Kuete, V. (2013). Antidiabetic activity of African Medicinal Plants. Medicinal Plant Research in Africa, 753–786. doi:10.1016/b978-0-12-405927-6.00020-5
  • Modak, M., Dixit, P., Londhe, J., Ghaskadbi, S., & Devasagayam, T. P. (2007). Indian herbs and herbal drugs used for the treatment of diabetes. Journal of clinical biochemistry and nutrition, 40(3), 163–173. https://doi.org/10.3164/jcbn.40.163
  • Atangwho, I. J., Yin, K. B., Umar, M. I., Ahmad, M., & Asmawi, M. Z. (2014). Vernonia amygdalina simultaneously suppresses gluconeogenesis and potentiates glucose oxidation via the pentose phosphate pathway in streptozotocin-induced diabetic rats. BMC complementary and alternative medicine, 14, 426. https://doi.org/10.1186/1472-6882-14-426
  • Gurib-Fakim A. Medicinal plants: traditions of yesterday and drugs of tomorrow. Mol Aspects Med. 2006 Feb;27(1):1-93. doi: 10.1016/j.mam.2005.07.008. Epub 2005 Aug 18. PMID: 16105678.
  • Newman DJ, Cragg GM. Natural Products as Sources of New Drugs over the Nearly Four Decades from 01/1981 to 09/2019. J Nat Prod. 2020 Mar 27;83(3):770-803. doi: 10.1021/acs.jnatprod.9b01285. Epub 2020 Mar 12. PMID: 32162523.
  • Lin, Y., Shi, R., Wang, X., & Shen, H. M. (2008). Luteolin, a flavonoid with potential for cancer prevention and therapy. Current cancer drug targets, 8(7), 634–646. https://doi.org/10.2174/156800908786241050
  1. The journal "Pharmaceutics" requires a short abstract of about 200 words at maximum. Following the suggestions of the Reviewers and the editorial request, the abstract has been modified in order to make it more relevant to the research performed.
  2. There are certainly many plants that have been suggested for the treatment of diabetes mellitus, as the Referee points out. The review by Jacob & Narendhirakannan (2019) indicates that there are more than 400 experimentally proven medicinal plants with antidiabetic properties, but the complete mechanism has been investigated only for a hundred of them. An ongoing challenge is to deepen our knowledge of phytoconstituents and the mechanisms responsible for the antidiabetic activity, and this was the objective that moved the present study, focusing on Vernonia amygdalina. Furthermore, an author of this research is from the African continent (Cameroon), so she knows very well the traditional use of Vernonia amygdalina. As explained in the introduction of the manuscript, the role of the two selected compounds, luteolin and vernodalol, has been investigated with respect to antioxidant, anti-α-glucosidase and antiglycant activities, all of which are of potential interest in the treatment and prevention of diabetes mellitus. Additionally, to study the selected compounds as possible candidates to find new antidiabetic agents, their effects on cell viability was evaluated, as well as in silico ADMET properties. Additional specifications have been incorporated into the manuscript.
  • Jacob B, & Narendhirakannan R T (2019). Role of medicinal plants in the treatment of diabetes mellitus: a review. 3 Biotech, 9(1), 4. https://doi.org/10.1007/s13205-018-1528-0
  1. According to the main purpose of the investigation, the assays considered have been selected from the documented and available ones to investigate the antioxidant, anti-α-glucosidase and antiglycant activities of the two significant phytoconstituents of Vernonia amygdalina. It was not a screening approach, but a targeted investigation of the main mechanisms to explain the antidiabetic properties attributed to the plant, to rationalize the traditional use and pose the basis for a possible drug development.
  2. Antioxidant activity is a new way to deal with complications of diabetes mellitus (Su et al. 2022). Increased free radical production and increased oxidative stress are well-known recognized factors affecting the morbidity and mortality of diabetes mellitos (Ceriello et al., 2016). Therefore, it is essential to focus on inhibitors to study new aspects of diabetes treatment that are not related to direct hyperglycemia control. For example, the presence of an antioxidant activity identified in the mechanisms of action of current antidiabetic drugs, such as metformin, glibenclamide, and repaglinide (Chukwunonso Obi et al., 2016), represents a new horizon that merits in-depth analysis and cannot be ignored.
  • Meiming Su, Wenqi Zhao, Suowen Xu, Jianping Weng. Resveratrol in Treating Diabetes and Its Cardiovascular Complications: A Review of Its Mechanisms of Action. Antioxidants 2022, 11(6), 1085; https://doi.org/10.3390/antiox11061085
  • Ceriello, A., Testa, R., & Genovese, S. (2016). Clinical implications of oxidative stress and the potential role of natural antioxidants in diabetic vascular complications. Nutrition, Metabolism and Cardiovascular Diseases, 26(4), 285-292.
  • Chukwunonso Obi B, Chinwuba Okoye T, Okpashi VE, Nonye Igwe C, Olisah Alumanah E. Comparative Study of the Antioxidant Effects of Metformin, Glibenclamide, and Repaglinide in Alloxan-Induced Diabetic Rats. J Diabetes Res. 2016;2016:1635361. doi: 10.1155/2016/1635361. Epub 2015 Dec 28. PMID: 26824037; PMCID: PMC4707348.
  1. The authors agree that the term "biomarker" is not appropriate in this context; therefore, following the suggestion it has been replaced by “active compound”, also according to the conclusions that indicate “Luteolin and vernodalol can be proposed as reference compounds that can help standardize medicinal preparations”.
  2. In silico ADMET have been studied with the aim of providing an exploratory view on the pharmacokinetics and toxicity that can guide the development of a new drug, determining, in a preliminary way, whether the compound is suitable for human use, including when administered orally, according to the current survey methodology (Gleeson et al., 2011; Cheng et al., 2013). Of course, the parameters of luteolin and vernodalol are independent of their plant sources; the data are presented as a general pharmacological profile of individual compounds, regardless of their origin. Otherwise, even in the presence of a suitable pharmacodynamic profile, a compound without favorable ADMET characteristics hardly will become a drug candidate.
  • Gleeson, M. P., Hersey, A., Montanari, D., & Overington, J. (2011). Probing the links between in vitro potency, ADMET, and physicochemical parameters. Nature reviews. Drug discovery, 10(3), 197–208. https://doi.org/10.1038/nrd3367
  • Cheng F, Li W, Liu G, Tang Y. In silico ADMET prediction: recent advances, current challenges and future trends. Curr Top Med Chem. 2013;13(11):1273-89. doi: 10.2174/15680266113139990033. PMID: 23675935.
  1. The authors regret that the figures appear to be of low quality, but the originals provided are in high resolution. The blurred aspect is due to the automatic conversion to pdf of the original files operated by the online submission system. In particular, Figure 1 was drawn using a specific chemical structure software (PubChem Sketcher V2.4) and is not a screenshot.
  2. The colors of the bars in the figures have been chosen to make easier to distinguish the data in the different treatments. If this is a problem with visibility, they can be changed to black and white.
  3. The manuscript was written choosing to combine the Results with the Discussion, and then Conclusions. Therefore, the conclusions are relatively short. Consistent with the suggestions of also the other Reviewers, the conclusions were not changed, while the Results and Discussion sections were extended. Thank you for your helpful suggestion to improve the manuscript.
  4. The manuscript has been improved to clarify the novelty of this search. If some data on luteolin are in part already known, the data on vernodalol are quite new since very few manuscripts have been published on this compound and, moreover, none on its action against alpha-glucosidase activity and protein glycation.
  5. MIUR (or MUR) is an acronym for the Italian “Ministero dell'Università e della Ricerca”, which means “Ministry of University and Research”, an institution that does not have any commercial interest, but exclusively supports independent research in Italian Universities. The explanation of the acronym has been added to the manuscript.
  6. This research does not serve any commercial purpose. But it wants to be a contribution to scientifically support the use of a medicinal plant traditionally used in Africa and Asia. The authors also want to find natural compounds, also common in foods, studying their potential utility against diabetes mellitus. The manuscript was improved by taking into account the suggestions, also excluding any commercial purpose.
  7. Vernonia amygdalina leaves are currently used in many African regions as a food (Alara et al., 2017), in addition to the medicinal properties. This provides additional relevance in supporting the use of this plant, as it can provide nourishment and, at the same time, act as a prevention of the progression of metabolic diseases, such as diabetes mellitus.

Round 2

Reviewer 3 Report

This is the revised version of amnsucript reproting on the activity of luteolin and vernodalol. HOwever, the rebuttal letter is in parts quite informative. Unfortunately, authors missed to implement the well justifications inot the manuscript. Further, I did not realize the suggested extension of the results and discussion part.

Still:

1) Figure 1 still looks like copied....needs to be improved. This is not comprehensive for a scientific publication. All other authors of papers can do as well...(Here experienced co-authors or supervisors can at all)

2) Discussion needs more comparing and adding some of the quite well statements from the rebuttal letter.

3) In some parts the terem "biomarker" was still used. Simply forgotten to change and with a certain reason. I still feel that even there it does not really fit in meaning.

4) The patterns in the bars of the figures make those figures quite twitchy. Maybe those paterrns are not necessary when using the colors only....

5) As concerns were quite high in the first review round with larg number of comments, quite less was done in the revised text....

6) It still needs reasons for the selection of assays (as justification).

7) I do not really argue against publishig, but it needs to be more interesting for potential readers....

8) It is not about satisfying the reviewer, but to improve it for the scientific community to get at least some reads and citations (at the moment it is still too poor to get acceptance from a broader community).

Author Response

Authors’ answers

  • Figure 1. In the original manuscript, the structural formulas are from vectorial drawing (ACD/ChemSketch), and if some inadequate view occurs, this is due solely to the pdf automatic conversion of the submission system that does not correspond to the real Figure definition. All graphs have been updated using a higher definition (see the manuscript in Microsoft Word). For example, the chemical structures of luteolin and vernodalol are reported below, as in Figure 1 (pdf).

  • Additional comparison statements have been introduced in the manuscript.
  • Lines 79 and 196. The term “biomarkers” was revised using “reference compounds”.
  • The patterns in the Figure bars were used together with colours to help vision for people who have some difference in vision, for example, people with colour blindness. In any case, probably, the Figures can be better without the patterns. Thus, all the Figures have been revised using only the colours. The authors believe that, effectively, the Figures now appear clearer.
  • In the Results and Discussion and Conclusion sections, additional discussion has been provided as suggested by the reviewer.
  • The rationale for the tests selected in this research was presented at the end of the Introduction, before Figure 1.
  • The authors have tried with determination to improve the manuscript, as indicated by the reviewer.
  • The authors agree on the need to draw the attention of the scientific community, as wide as possible, and for this reason they have revised the manuscript, expanding the Results and Discussion, and Conclusion, as suggested by the reviewer.

Round 3

Reviewer 3 Report

No further comments....

Author Response

The authors thank the Reviewer for the comments that contributed to the enhancement of the manuscript.
